# Higher-order epistasis shapes natural variation in germ stem cell niche activity

**Sarah R. Fausett** [1,2] ✉, **Asma Sandjak**[1,3], **Bénédicte Billard**[1,3] & **Christian Braendle** [1] ✉

To study how natural allelic variation explains quantitative developmental system variation, we characterized natural differences in germ stem cell niche activity, measured as progenitor zone (PZ) size, between two *Caenorhabditis elegans* isolates. Linkage mapping yielded candidate loci on chromosomes II and V, and we found that the isolate with a smaller PZ size harbours a 148 bp promoter deletion in the Notch ligand, *lag-2/Delta*, a central signal promoting germ stem cell fate. As predicted, introducing this deletion into the isolate with a large PZ resulted in a smaller PZ size. Unexpectedly, restoring the deleted ancestral sequence in the isolate with a smaller PZ did not increase—but instead further reduced—PZ size. These seemingly contradictory phenotypic effects are explained by epistatic interactions between the *lag-2/Delta* promoter, the chromosome II locus, and additional background loci. These results provide first insights into the quantitative genetic architecture regulating an animal stem cell system.

Fine-tuning of cellular proliferation is a fundamental aspect of organismal development and tissue homeostasis, often coordinated by stem cell niches. Even small perturbations in stem cell niche activity can deregulate tissue growth and maintenance to cause pathologies[1]. Dissecting the molecular genetic mechanisms regulating the activity of stem cell niches has, therefore, become a major focus of biological research. While developmental genetic studies of stem cell niche function in animals have unravelled underlying key molecular regulatory mechanisms, whether and how activity of stem cell systems is modulated by genetic variation segregating in natural populations remains largely unaddressed. If it exists, how does such allelic variation contribute to variation in stem cell niche activity? Do known genes involved in stem cell niche signalling harbour this variation? And to what extent can natural variation in stem cell niche activity be explained by effects of single, large-effect genetic variants versus polygenic contributions of small-effect variants? Most quantitative traits are complex, involving a polygenic architecture, with genetic variants not only acting additively but also in an interactive manner. Such epistasis, also termed gene-gene (G x G) interactions, corresponds to non-additive interactions between allelic variants at different genomic loci[2]. Strong polygenicity and epistasis have been

observed for most quantitative phenotypes across divergent taxa[3–8], but detailed mechanistic dissection of complex epistatic interactions, including higher-order epistasis where three or more loci interact, remains rare[9–15]. Although experimentally difficult to characterize, molecular and quantitative genetic analyses also suggest that widespread epistatic interactions underlie developmental phenotypes[15–24]. Yet, so far, there is no information on how interactions among natural alleles cause quantitative variation in animal stem cell systems.

Germline stem cell (GSC) systems are fundamental to metazoan development and reproduction, maintaining immortal germ cell populations in an undifferentiated state and integrating genetic and environmental cues to adjust the production of germ cell progenitors[1,25,26]. Genetic research and comparative evo-devo studies have uncovered a diversity of GSC systems across distant taxa[27–29], but whether the activities of these systems show quantitative variation in natural populations of the same species is currently not known. Genomic and developmental genetic analysis of closely related species (e.g., within the genus *Drosophila* or the nematode genus *Caenorhabditis*) indicate that principal features, such as the key molecular signalling pathways and cell-cell interactions, of the GSC niche are largely conserved within genera[30–34]. Nevertheless, population-genetic

[1]Université Côte d'Azur, CNRS, Inserm, IBV, Nice, France. [2]Department of Biology and Marine Biology, University of North Carolina Wilmington, Wilmington, NC, USA. [3]These authors contributed equally: Asma Sandjak, Bénédicte Billard. ✉e-mail: fausetts@uncw.edu; braendle@unice.fr

studies in *Drosophila* show that central GSC genes can harbour surprisingly high levels of allelic variation, even within species, suggesting that these genes evolve rapidly and often due to positive selection[35–37]. Therefore, despite their importance in a fundamental developmental process, regulatory genes of GSC niches do not seem to be evolutionarily constrained. What remains unclear is how observed natural allelic variation translates into phenotypic variation, such as GSC niche activity.

To study the genetic basis of natural quantitative variation in GSC niche activity, we use the GSC system in *C. elegans*, which has served as a simple in vivo system to study stem cell niche function, involving a set of well-defined molecular signalling pathways[25,38,39]. The *C. elegans* hermaphrodite germline consists of two symmetrical arms with distal germ cell progenitor zones (PZ), also termed mitotic or proliferative zones, that include the stem cells, as well as progenitor cells in mitosis and meiotic S phase (Fig. 1a). Germ cells differentiate into gamete progenitors through meiotic stages as they progress toward the proximal end of the arm (Fig. 1a, b)[25,38]. The key regulatory signals are expressed by the Distal Tip Cell (DTC), a somatic gonadal cell that caps and contacts cells in the PZ (Fig. 1a). These signals are the Delta/Serrate/LAG-2 (DSL-family) ligands, LAG-2 and APX-1, which activate Notch (GLP-1) receptors in distal germ cells, promoting the stem cell fate and inhibiting entry into meiosis[40–46]. The DTC thus constitutes a stem cell niche[25]. GLP-1/Notch signalling is necessary and sufficient for the maintenance of the germ stem cell pool[25,39]. Germ cells progressively enter meiosis as they move proximally and lose contact with the DTC. This is controlled by a network of RNA regulatory proteins, including PUF (Pumilio and FBF) RNA-binding proteins that promote self-renewal downstream GLP-1/Notch and other RNA regulatory proteins (GLD-1, GLD-2, SCF$^{PROM-1}$) that promote entry into meiosis[25,38,39,47–49]. Additional signals from gonadal sheath cells further modulate *C. elegans* germ cell proliferation and differentiation[50–56]. Germline PZ cell number is thus determined by the interplay of distal proliferative activity and the spatio-temporal transition into the proximal meiotic state. Moreover, the proliferative activity of the *C. elegans* germline is highly sensitive to variation in physiology and the external environment, differing in response to nutrient quality and quantity, temperature, or social environment[25,32,57–61]. Environmental variation modulates GSC proliferation via metabolic and sensory signalling pathways (e.g., TGF-β, TOR, AMPK and insulin), which act both dependently and independently of niche-mediated Delta/Notch signaling[62–68]. The *C. elegans* germ stem cell system is thus highly plastic, capable of fine-tuning its activity in response to subtle environmental changes. Whether natural genetic variation can similarly modulate the activity of this stem cell system is not known. However, previous reports suggest that the size of the germline progenitor pool varies not only among different *Caenorhabditis* species but also among distinct *C. elegans* wild isolates[32,69].

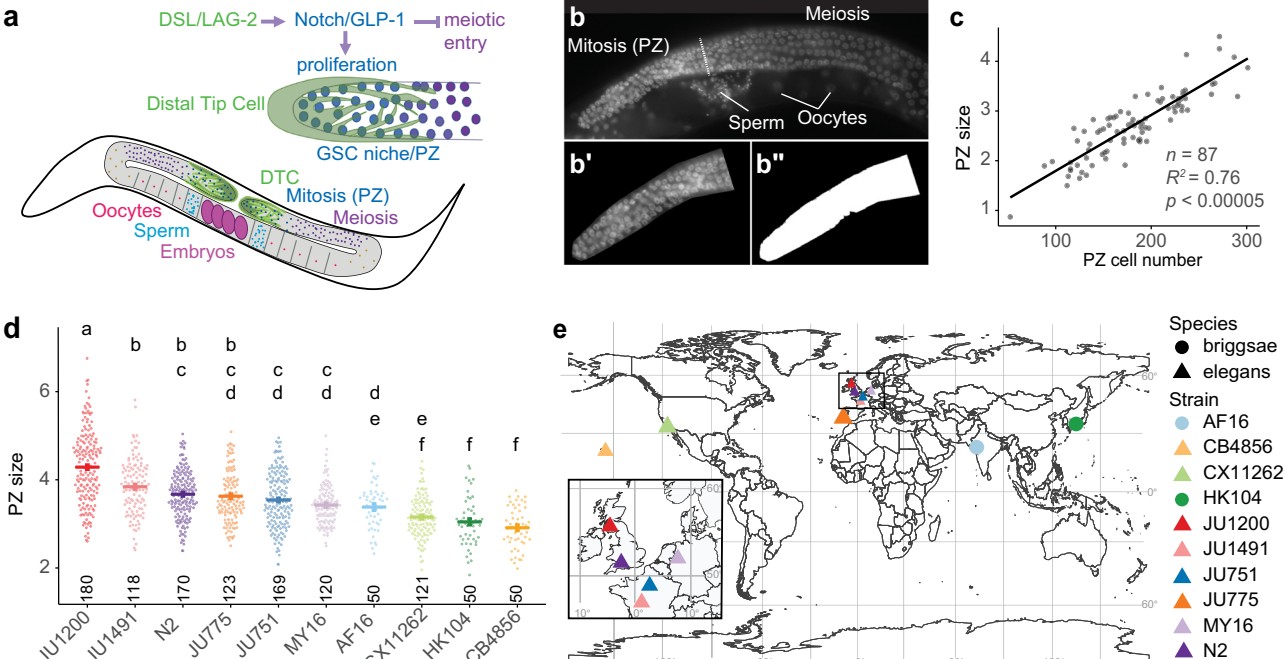

**Fig. 1 | Germline progenitor zone size varies among natural *Caenorhabditis* populations. a** The germline progenitor zone (PZ), located at the distal end of each gonad arm, contains mitotically dividing stem and progenitor cells. The germ stem cell (GSC) pool is maintained through Delta/Notch signalling by the somatic Distal Tip Cell (DTC), which enwraps distal PZ cells. The Delta/Serrate/LAG-2 (DSL-family) ligands, LAG-2 and APX-1, activate Notch (GLP-1) in germ cells to promote proliferation while also preventing meiotic entry. Germ cells differentiate into gamete progenitors through meiotic stages as they progress toward the proximal end of the arm. **b** PZ size was used as a proxy for PZ cell number. Whole-mount worms stained with DAPI were imaged under epifluorescence. The PZ boundary (dotted line) was identified as previously described based on germ cell nuclei shape[38]. **b'** The PZ region was manually cropped away from other tissues in ImageJ. **b"** The PZ region was segmented from the background using a fluorescence threshold. The PZ area was measured in pixels. Scale bars 20μm. **c** PZ area (scaled to PZ size–μm²/ 1000) measured this way correlates well with hand-counted PZ cell number. Data were obtained from measurements of the two isolates JU1200 and JU751 at two adult stages: mid-L4 + 24 h and young adult + 24 h. Each data point represents an individual (linear model adj $R^2$ = 0.758, $p$ < 2.2E-16, $n$ = 87); the experiment was repeated twice with similar results. **d** PZ size (μm²/1000) in young adult hermaphrodites (1-10 eggs *in utero*) of select wild isolates of *C. elegans* and *C. briggsae*. Cross bars and error bars represent estimated marginal means ± standard errors derived from a generalized linear model. Lowercase letters indicate significant ($p$ < 0.05) Tukey-adjusted pairwise contrasts: isolates that share the same letters do not exhibit a significant difference in PZ size. *n*-values across two blocks are indicated above the x-axis. **e** Geographical origins of examined wild isolates; maps produced using the R packages rnaturalearth and rnaturalearthdata v. 0.1.0[112] and rgeos v. 0.5-9[113]. For data and statistical results, see Supplementary Notes 1 and 2 and the Source Data file.

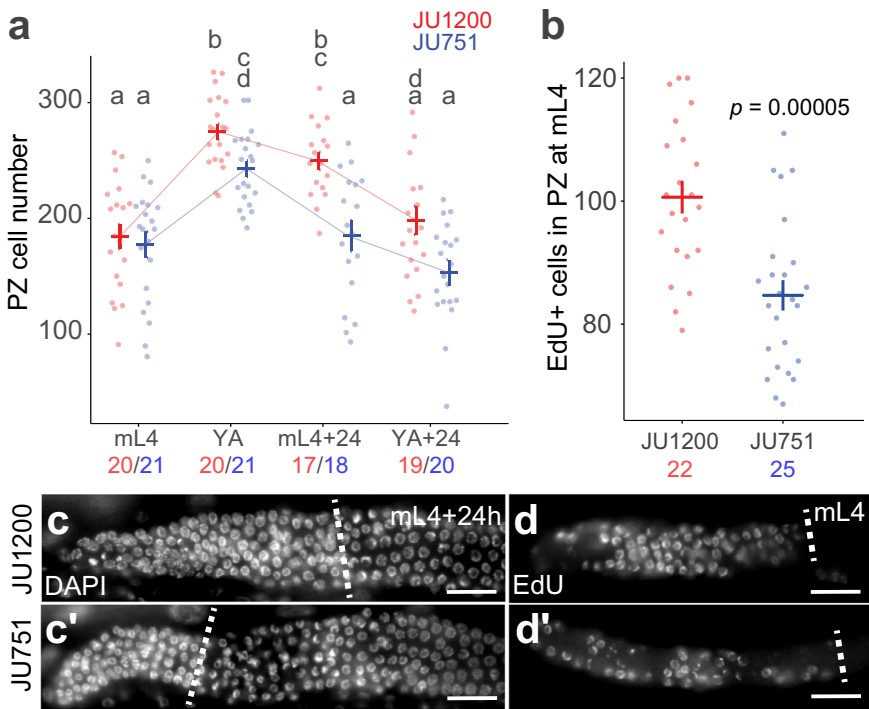

**Fig. 2 | Germ cell proliferative activity differs between wild isolates JU1200 and JU751. a** Number of PZ nuclei at four developmental stages: mid-L4 larval (mL4), young adult (YA) with 1-10 eggs *in utero*, adult at 24 hours post-mid L4 (mL4 + 24 h), and adult at 24 hours post-young adult (YA + 24 h). Nuclei were counted in z-stacks of extruded DAPI-stained gonads. Lowercase letters indicate significant Tukey-adjusted pairwise contrasts (*p* < 0.05), so that groups sharing the same letters are not significantly different. Cross bars and error bars represent estimated marginal means ± standard errors derived from a generalized linear model. *n*-values for each strain and stage are indicated below the x-axis (JU1200=red, JU751=blue) **b** Number

of EdU+ (positive) nuclei in the PZ after a 15-minute pulse of EdU at the mid-L4 stage (***p = 0.00005, Tukey-adjusted pairwise contrast). *n*-values are indicated below the x-axis. Cross bars and error bars represent estimated marginal means ± standard errors derived from a generalized linear model. **c** Representative images of the PZ in whole-mount DAPI-stained worms at mL4 + 24 h. Scale bars: 20μm. **d** Representative images of EdU staining at the mid-L4 stage. Dotted lines mark the proximal boundary of the PZ. Scale bars: 20μm. Data are shown from a single (out of two) experimental blocks. For data and statistical results, see Supplementary Notes 3 and 4 and the Source Data file.

In this study, we aimed to quantify and genetically characterize natural variation in *C. elegans* germ stem cell niche activity. We show that germline progenitor zone (PZ) size—here, a proxy for germ stem cell niche activity—differs between genetically and geographically distinct wild isolates under identical standard lab conditions. We used a linkage mapping approach to identify genomic regions associated with natural variation in PZ size, concentrating on two wild isolates with pronounced differences in PZ size. Genetic mapping revealed four candidate QTL including two large-effect loci on chromosomes II and V that act additively to explain ~32% of the variation in PZ size among the lines of our mapping panel. We discovered that the QTL region on chromosome V harbours an INDEL variant in the promoter region of the DSL ligand *lag-2*, which causally contributes to variation in PZ size. However, we also discovered that the phenotypic effects of this variant are strongly modulated through epistatic interactions with the QTL on chromosome II and additional, unknown genomic loci. Taken together, our results indicate that complex epistatic interactions are important contributors to natural variation in germ stem cell niche activity and provide first insights into the quantitative genetic architecture of an animal stem cell system.

## Results

### Caenorhabditis wild isolates differ in PZ size
Under standard laboratory conditions, the hermaphrodite germline progenitor zone (PZ) of the reference wildtype *C. elegans* strain (N2) typically contains ~250 cells during early adulthood[25]. Imaging through the entire PZ and counting the total number of germ cells is time-consuming and inefficient for screening large numbers of strains. To quantify natural differences in PZ size between multiple wild isolates of

*Caenorhabditis*, we first developed a method to approximate PZ size by quantifying the trans-sectional area of the PZ in a single fluorescent image of a whole-mount DAPI-stained gonad (Fig. 1b). PZ measurements derived by this method correlated well with individual counts of PZ nuclei from image stacks through the entire PZ (adj. $R^2$ = 0.758, *p* < 2.2E-16) (Fig. 1c, Supplementary Note 1). We examined several wild isolates of *C. elegans* and *C. briggsae* from around the globe and discovered significant inter- and intraspecific variation in PZ size when measured in standard lab conditions (Fig. 1d, Supplementary Note 2). The broad-sense heritability of young adult PZ size in the data set was estimated to be 28% (Supplementary Note 2d).

### Germ cell proliferation differs between JU1200 and JU751
Among the examined wild isolates, JU1200 (Scotland) and JU751 (France) exhibited significant and highly reproducible differences. JU1200 shows high genetic similarity with the laboratory reference strain N2[70]. As young adult hermaphrodites, JU1200 exhibited consistently more progenitor cells than JU751 (Fig. 1d). We assayed PZ cell number in these two isolates from late larval to reproductive adult stages. While PZ cell number did not differ at the mid-L4 stage, JU751 showed significantly reduced PZ cell number relative to JU1200 across two stages of early adulthood (Fig. 2a, Supplementary Note 3).

Given that PZ size may not only be affected by proliferative activity but also by the transition rate towards the meiotic fate, we assayed mitotic activity using a 15-minute EdU (5-ethynyl-2-deoxyuridine) pulse to directly label and quantify proliferation in the PZ prior to the adult stage. At the mid-L4 stage, when PZ cell number in JU751 and JU1200 were not significantly different (Fig. 2a), JU751 exhibited significantly reduced counts of EdU-positive cells, and thus lower germ

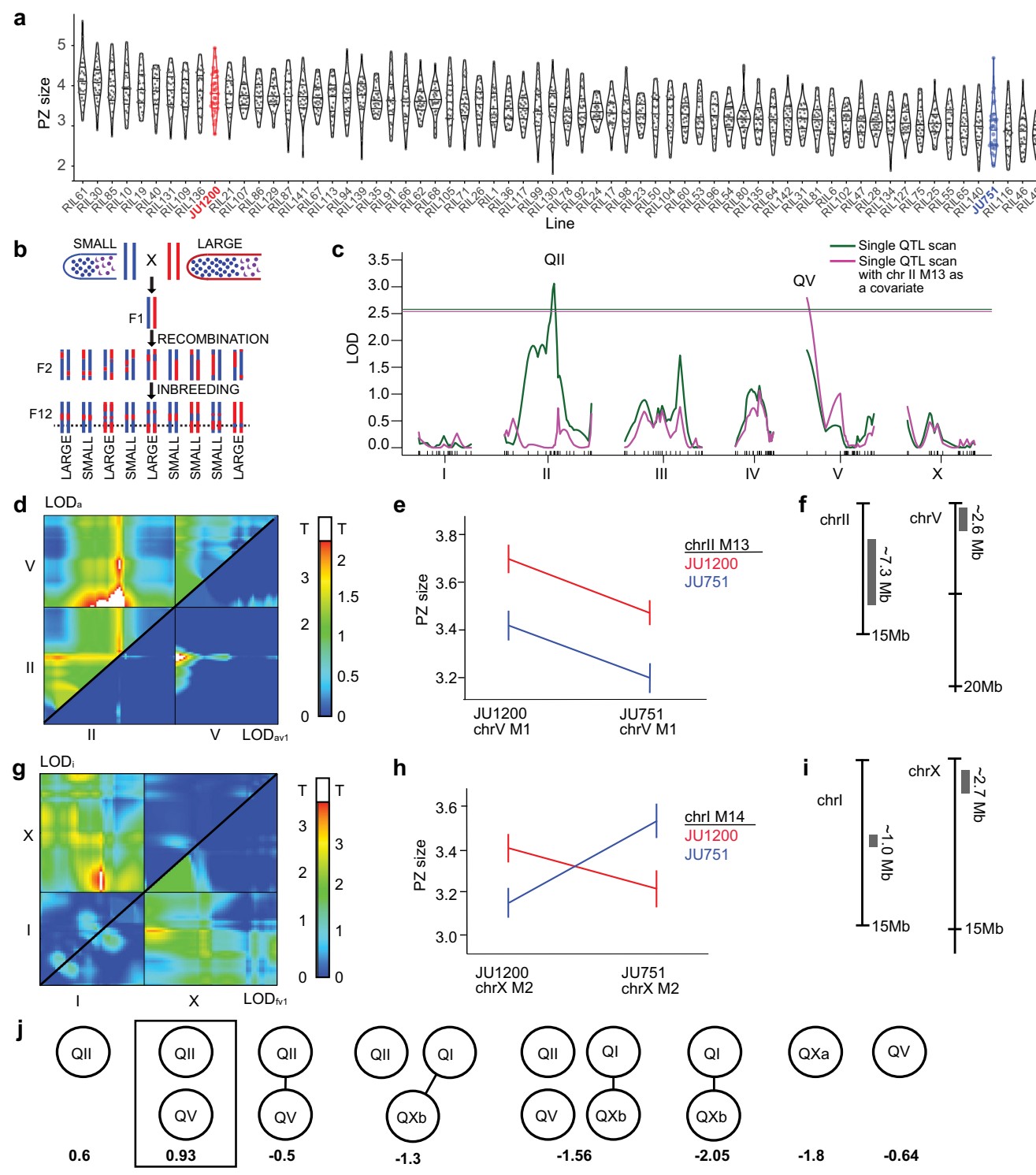

cell proliferative activity than JU1200 (Fig. 2b, Supplementary Note 4). Differences in germ cell proliferation thus contribute to the observed difference in PZ size between JU751 and JU1200.

**Multiple loci contribute to natural variation in PZ size**

To characterize the genetic architecture underlying the observed differences in PZ size between JU751 and JU1200, we performed Quantitative Trait Locus (QTL) linkage mapping using existing F2 recombinant inbred lines (RILs) derived from these two isolates (Fig. 3a, b)[71]. The RILs were constructed by selfing F1 hermaphrodites from reciprocal parental crosses and inbreeding the F2 lines for twelve

generations to produce a panel of lines homozygous at each locus for one of the two parental genotypes (Fig. 3b)[71]. We measured the PZ size of young adult hermaphrodites (1-10 eggs *in utero*) in 70 RILs and mapped the phenotypic differences to a genetic map derived from the recombination frequencies of 140 SNP markers throughout the genome (Fig. 3a, Supplementary Fig. 1). PZ size varied continuously across RILs and showed transgressive segregation (i.e., extreme phenotypes exceeding the parental phenotypes) (Fig. 3a). We, therefore, employed a multi-QTL modelling approach using the R package, R/qtl[72]. As a first step, we performed a single QTL scan, an algorithm that calculates the likelihood that the genotype at any single locus can explain the

**Fig. 3 | Linkage mapping identifies multiple QTL explaining variation in PZ size.**
**a** PZ size estimates (corrected) in F2 RILs (n = 70, assayed across six experimental blocks) and parental lines, JU1200 (red), and JU751 (blue) (see Supplementary Note 5 for details). **b** Generation and analysis of F2 RILs. Dotted line: hypothetical QTL. **c** Single-QTL scans identify two QTL (QII and QV). Horizontal lines: LOD threshold based on permutation tests. x-axis: genetic map for each chromosome. Green line: naïve scan results. Magenta line: results of a scan where marker M13 (peak, magenta curve) is included as a covariate. **d** Two-QTL scan results. Top half: LOD scores (LOD$_a$) comparing additive to null model. Bottom half: LOD scores comparing additive to single-QTL model. LOD$_{av1}$ scores above threshold (white, determined by permutation testing) indicate improvement in fit over single-QTL model. **e** Interaction plot for markers in the peak in panel D, bottom half (chr V, M1; chr II, M14). **f** Approximate locations and sizes of the QTL on II and V. **g** Two-QTL scan allowing opposing effects. Top half: LOD scores (LOD$_i$) comparing a full two-QTL model to the additive model. Bottom half: LOD scores comparing the full model (allowing for interactions) with the single-QTL model (LOD$_{fv1}$). LOD$_i$ scores above threshold (white) indicate evidence for an interaction, whereas LOD$_{fv1}$ scores above the threshold indicate an improvement in fit over the single-QTL model. **h** Interaction plot for markers in the significant peak in the top half of panel G (chr I, M14 and chr X, M2). **i** Approximate locations and sizes of candidate QTL on chromosomes I and X. **j** Representation of multi-QTL model selection. Nodes represent candidate QTL (QII = chrII@35 cM, QV = chr V@0 cM, QI = chrI@23 cM, QXa = chrX@16 cM, QXb = chrX@2.5 cM) and spokes represent interactions. Penalized LOD scores (below each model) above zero indicate better performance than the global null (zero QTL). See Methods, Supplementary Fig. 1, and Supplementary Note 6 for more detail. Y-axes in panels **a**, **e**, and **h** are scaled to µm²/1000 (0.1008 µm²/pixel). Data are provided as a Source Data file.

phenotype data. This scan revealed a large-effect locus in the centre of chromosome II (QII) with a 1.5-LOD interval of ~7.25 Mb (Fig. 3c, green line). Next, we used a two-QTL algorithm to determine the likelihood that any *pair* of loci is associated with the phenotype. Importantly, this algorithm does not allow the loci to have opposing effects. The two-QTL scan revealed an additional locus (QV) on the left arm of chromosome V (~2.6 Mb) acting additively with the QII QTL on chromosome II (Fig. 3d–f). Single-QTL mapping with the QII QTL as an additive covariate also revealed the QV QTL on chromosome V (Fig. 3c, magenta line). An additional two-QTL scan, specifically allowing for interactions between loci with opposing effects, revealed two more potential QTL on chromosomes I (QI, ~1 Mb) and X (QX, ~2.7 Mb) (Fig. 3g–i). We then performed multi-QTL model selection, using these four loci as candidates. We performed the model selection manually and also used a model-building and pruning tool provided in R/qtl. This algorithm begins naïvely or with a specified model, adding loci through successive genome scans and assigning each model a penalized LOD score. It then prunes the model to achieve the simplest form with the highest penalized LOD score (LOD > 0 indicates that the model performs better than a model with zero QTL). A representative set of all the models tested, along with their penalized LOD scores, are shown in Fig. 3j. We used both approaches and determined that our RIL data best support a model in which QII and QV act additively to determine PZ size. This model explained ~32% of the phenotypic variation in the RIL data (Supplementary Note 6).

## Near-isogenic lines validate the chromosome II QTL

To validate the effect of the chromosome II QTL (QII) on PZ size, we generated near-isogenic lines (NILs). Two RILs containing JU751 sequence in QII were backcrossed for 10 generations to JU1200 to produce two NILs containing JU1200 sequence in various segments of the QTL (Fig. 4a). Two RILs containing JU1200 sequence in QII were backcrossed for 10 generations to JU751 to produce five NILs containing JU1200 sequence in various segments of the QTL (Fig. 4b). NILs were generated by selecting for the presence of parent-specific PCR products in the QTL region. We assayed PZ size several times in each of the NILs. The NILs established in the JU1200 background showed subtle effects on PZ size, which were not always consistent across experimental assays (blocks). We, therefore, analysed the aggregated results obtained from nine blocks using generalized linear models to account for the experimental block effect and found that we could detect a small, but significant, reduction in PZ size (Fig. 4c, Supplementary Note 7). The NILs in the JU751 background showed larger and more consistent differences in PZ size, confirming the effect of QII on PZ size (Fig. 4d, Supplementary Note 8). Moreover, recombination events that had occurred during the backcrossing procedure allowed us to narrow down the QTL region from 7.25 Mb to 2.04 Mb (Fig. 4b). Searching for JU751-JU1200 polymorphisms in this reduced genomic region uncovered a total of ~1024 variants (including 196 potential high-impact variants, i.e., variants predicted by a BCSQ algorithm to

likely alter gene function, for example, through frameshift or nonsense mutations) in 250 distinct genes and 10 large INDELs[70] (see Source Data for details). However, we did not identify any strong candidate variants for further analysis.

## A deletion in the lag-2 promotor is a candidate causal variant

The 2.6 Mb region of QV contained only 21 presumptive candidate variants (SNPs and INDELS) contributing to PZ size differences between JU751 and JU1200. Inspecting these variants in detail revealed one major candidate: a deletion in JU751 upstream of the genomic region encoding the delta-like ligand, *lag-2*, a primary regulator of germ cell proliferation and differentiation in the Notch pathway[25,70]. We named this deletion *lag-2(cgb1007);* genotypes with this deletion are abbreviated from here forward as *lag-2p(-)*, whereas we refer to genotypes without this deletion as *lag-2p(+)*. This deletion spans 148 bp and is located in the *lag-2* promoter region, 2,441 bp upstream of the transcription initiation site (Fig. 5a and Supplementary Fig. 2). The 3.3 kb upstream promotor region of *lag-2* contains six HLH-2 binding sites (E-boxes), which are necessary for robust *lag-2* transcription in the DTC[30,73]. The *lag-2(cgb1007)* deletion contains one of these conserved HLH-2 binding sites as well as most of a 30 bp poly(G) repeat (Fig. 5a, Supplementary Fig. 2 and 3). This deletion could explain reduced germ cell proliferation observed in JU751 due to reduced *lag-2* transcription via the loss of a specific binding site for the transcription factor HLH-2[73].

Examining whole-genome sequence data of a global panel of over 1500 *C. elegans* wild isolates[70,74], we found the *lag-2(cgb1007)* deletion to be unique to the isolate JU751. A similar, but larger, deletion in the *lag-2* promoter region was found in two isolates from Asia, GXW1 and JU4073 (Supplementary Fig. 3c, d). Since no other isolates were found to carry the *lag-2(cgb1007)* deletion, we wondered if this deletion could have arisen after isolation, during laboratory culture. We, therefore, examined additional isolates (for which no whole-genome sequence data exist) that were collected from the same habitat (compost) and location (Le Perreux-sur-Marne, France) as JU751 in 2004 and 2005[75]. All other isolates (n = 5) that were collected together with JU751 from this locality on the same day shared a highly similar haplotype and also carried the *lag-2(cgb1007)* deletion. In contrast, all examined isolates with different haplotypes, collected at other dates throughout 2004 and 2005 (n = 13) did not carry the deletion[71] (Supplementary Fig. 3). We conclude that the *lag-2(cgb1007)* deletion was acquired before isolation in the laboratory and that this allele is likely rare in natural populations.

## The lag-2 variant has background-dependent effects on PZ size

To validate that the *lag-2(cgb1007)* genomic region contributes to differences in germline PZ between JU751 and JU1200, we created reciprocal allelic replacement lines (ARLs) using CRISPR-*Cas9* genome editing. First, we introduced the *lag-2(cgb1007)* deletion into JU1200, resulting in the strain JU1200$_{lag-2p(-)}$. Second, we restored the deleted

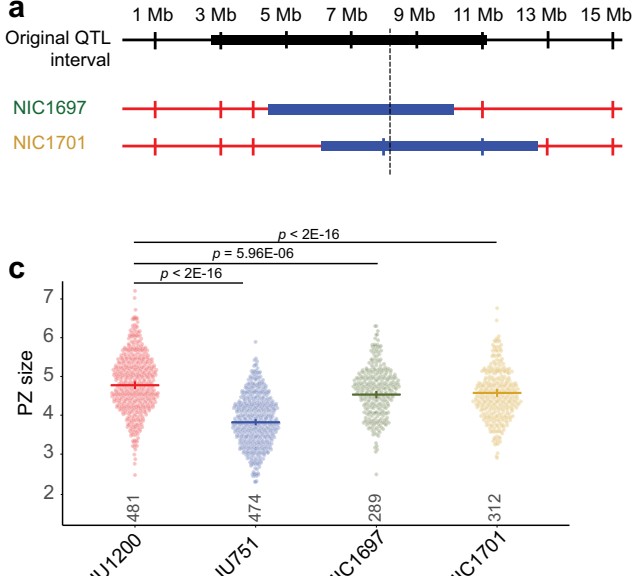

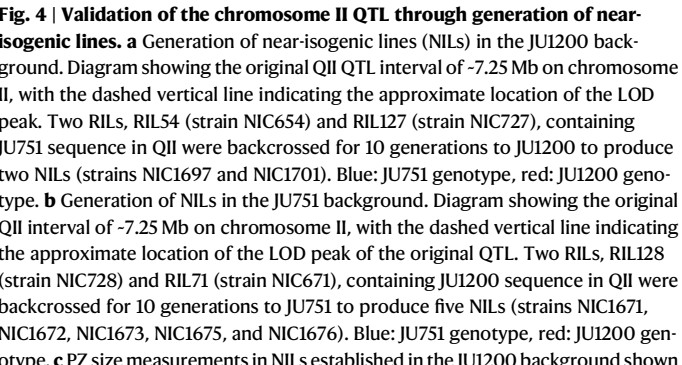

**Fig. 4 | Validation of the chromosome II QTL through generation of near-isogenic lines. a** Generation of near-isogenic lines (NILs) in the JU1200 background. Diagram showing the original QII QTL interval of ~7.25 Mb on chromosome II, with the dashed vertical line indicating the approximate location of the LOD peak. Two RILs, RIL54 (strain NIC654) and RIL127 (strain NIC727), containing JU751 sequence in QII were backcrossed for 10 generations to JU1200 to produce two NILs (strains NIC1697 and NIC1701). Blue: JU751 genotype, red: JU1200 genotype. **b** Generation of NILs in the JU751 background. Diagram showing the original QII interval of ~7.25 Mb on chromosome II, with the dashed vertical line indicating the approximate location of the LOD peak of the original QTL. Two RILs, RIL128 (strain NIC728) and RIL71 (strain NIC671), containing JU1200 sequence in QII were backcrossed for 10 generations to JU751 to produce five NILs (strains NIC1671, NIC1672, NIC1673, NIC1675, and NIC1676). Blue: JU751 genotype, red: JU1200 genotype. **c** PZ size measurements in NILs established in the JU1200 background shown

in panel **a** and the parental isolates. *n*-values are shown above the x-axis. Data from nine experimental blocks were analysed. **d** PZ size measurements in NILs established in the JU751 background shown in panel **b** and the parental isolates. *n*-values are shown above the x-axis. Data are shown from a single (out of nine) experimental blocks The reduced QII interval was determined by asking whether each NIL showed a significant increase in PZ size as compared to JU751. All NILs except NIC1676 had larger PZs than JU751. Therefore, the region over which the JU1200 segment in these NILs overlap was considered the reduced QTL interval (2.04 Mb). Analyses for data shown in panels **c** and **d** was carried out on raw data (PZ area in pixels), and the y-axes were scaled to $\mu m^2/1000$ for presentation ($0.0504\ \mu m^2$/pixel). Cross bars and error bars represent estimated marginal means ± standard errors derived from generalized linear models. *p*-values are derived from Tukey-adjusted pairwise contrasts. For data and statistical results, see Supplementary Notes 7 and 8 and the Source Data file.

*lag-2* promoter region in JU751, resulting in the strain JU751*lag-2p(+)*. As predicted, removing the 148 bp fragment in the JU1200 background strongly reduced PZ size (Fig. 5b). In contrast, and contrary to expectations, restoring the corresponding fragment in the JU751 background did not increase but further reduced PZ size (Fig. 5b, Supplementary Note 9). Although the *lag-2(cgb1007)* deletion is sufficient to decrease PZ size to a similar extent as in JU751 when introduced in the JU1200 background, it cannot be the only genetic variant contributing to the difference of PZ size observed between JU751 and JU1200. In other words, the effects of the *lag-2(cgb1007)* deletion are strongly dependent on the genetic background. This is in line with our QTL analysis, in which the marginal effect of the QV QTL was only detectable when accounting for variation at the QII QTL (Fig. 3c). Furthermore, while *lag-2(cgb1007)* has clear effects on PZ size, and therefore contributes to the overall effect of QV, we cannot rule out other causal variants within the QV QTL region.

### lag-2 expression matches genotypic differences in PZ size
Given the result above, we next wanted to directly determine whether and how observed genotypic differences in PZ size can be explained by differences in *lag-2* expression. Using single-molecule fluorescence in situ hybridization (smFISH)[76,77], we quantified *lag-2* transcripts in the DTCs of the parental isolates and reciprocal ARLs, JU1200*lag-2p(-)* and JU751*lag-2p(+)*. We first measured *lag-2* expression in parental isolates and found that expression was considerably higher during larval development than in the adult stage, with the strongest expression during the early L3 stage (Supplementary Fig. 4, Supplementary Note 12). Quantifying *lag-2* expression in parental isolates and

reciprocal ARLs across three larval stages, we found that at the early L3 stage (but not at mid-L2 or at mid-L4), JU1200 showed significantly higher *lag-2* expression relative to JU751 (Fig. 5d, Supplementary Note 10), consistent with its larger PZ size. Remarkably, relatively higher expression of *lag-2* in JU1200 at the L3 stage was completely abolished in JU1200*lag-2p(-)*, which showed very similar expression levels as JU751 across all three developmental stages (Fig. 5e, Supplementary Note 10). These results are consistent with past results showing that the *lag-2* promotor E-boxes are required for robust *lag-2*::GFP transgene expression during the L3 stage[73]. Inserting the 148 bp *lag-2* promoter sequence into the JU751 background did not result in higher *lag-2* expression at L3 relative to JU751. Instead, *lag-2* expression in JU751*lag-2p(+)* was lower at L3 with similar trends in expression changes among the stages (Fig. 5f, Supplementary Note 10). This result confirmed that the unexpected reduction in PZ size of JU751*lag-2p(+)* is indeed causally linked to reduced *lag-2* expression. Overall, these smFISH measurements show that *lag-2* expression (at the L3 stage) tightly recapitulates observed phenotypic differences in PZ size at the early adult stage and confirm that the effects of the *lag-2(cgb1007)* deletion are strongly dependent on the genetic background. Analysis of the parental isolates also shows that natural differences in *C. elegans* germ stem cell niche activity can be directly linked to differences in the expression of a core signalling component, the Notch ligand *lag-2*.

### Higher-order epistasis shapes natural variation in PZ size
Given the apparent epistatic interactions between the *lag-2* promoter variant and the genetic backgrounds, we built a more exhaustive model to account for additional two- and three-way interactions

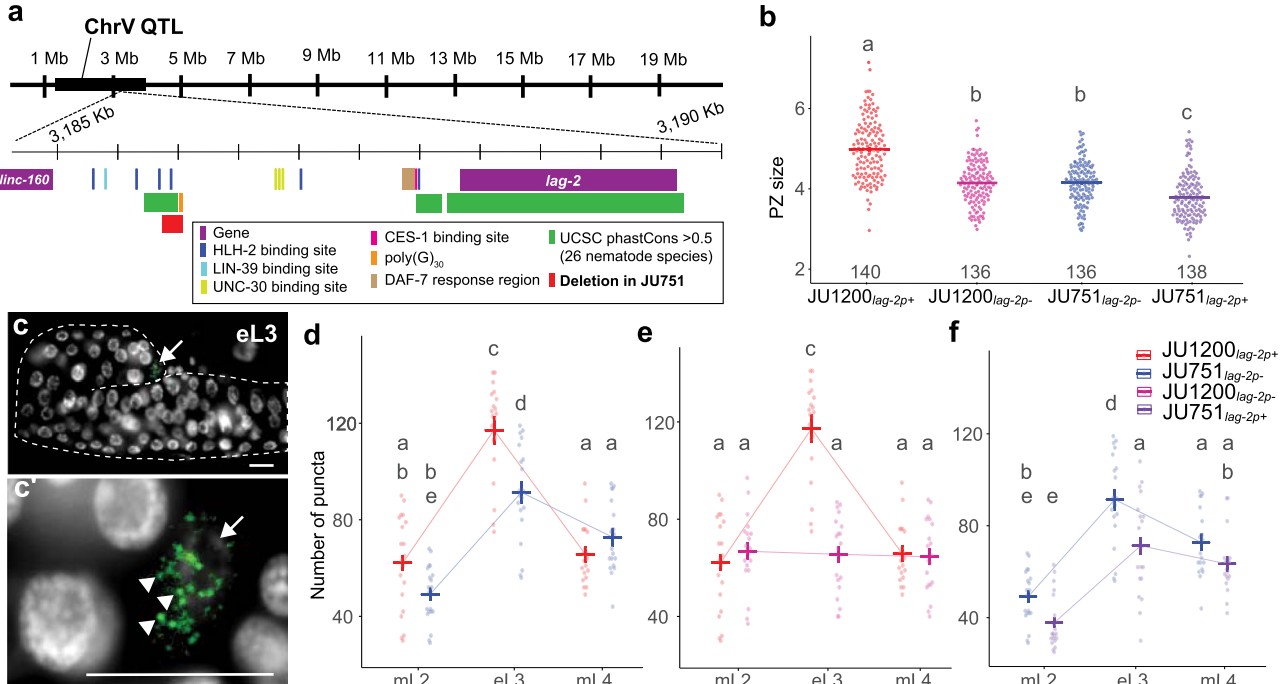

**Fig. 5 | A 148 bp deletion in the upstream regulatory region of the DSL ligand *lag-2* has background- and stage-specific effects on *lag-2* expression and PZ size. a** Diagram of the QV QTL (black bar) and a 5 kb region within it containing the *lag-2* gene and its 3 kb upstream sequence. The well-characterized *lag-2* promoter contains several known transcription factor binding sites including six HLH-2 E-boxes (dark blue)[115]. JU751 has a 148 bp deletion (red) overlapping the third HLH-2 binding site and most of a 30 bp polyG repeat (orange). The locations of all promoter elements and UCSC phastCons statistics (green) were taken from the wormbase.org (version WS283) JBrowse tool[115]. **b** The size of the PZ (μm²/1000) in the parents and allelic replacement lines (ARLs) at the young adult stage. Crossbars and error bars represent estimated marginal means ± standard errors from a generalized linear model. *n*-values across two blocks are shown above the x-axis. Analysis was carried out on raw data (PZ area in pixels), and the y-axes were scaled to μm²/1000 for presentation (0.0504 μm²/pixel). **c** Representative images used for smFISH quantification of *lag-2* transcripts in the DTC at mid L3 (*lag-2* mRNA green, DAPI [DNA] white, scale bars 10μm). The gonad arm is outlined by a dashed line. The arrow marks the DTC. **c'** Higher magnification of DTC area from **c**. Arrow indicates the DTC. Arrowheads indicate individual puncta, which were quantified to determine *lag-2* expression levels. **d–f** Quantification of *lag-2* mRNA puncta via smFISH. Crossbars and error bars represent estimated marginal means ± standard errors from a single generalized linear model. For all genotypes, *n* = 19 or 20 per stage. Data from a single experiment were split into three graphs for clarity. Groups with different lowercase letters are significantly different (*p* < 0.05) according to Tukey-adjusted pairwise contrasts. For data and statistical results, see Supplementary Notes 9 and 10 and the Source Data file.

among 1) central portions of the chromosome II QTL isolated in our NILs (abbreviated C2 to distinguish it from the full QII QTL—with genotypes *C2-JU1200* or *C2-JU751*), 2) the *lag-2* promoter variant (abbreviated C5—with genotypes *lag-2p(+)* or *lag-2p(-)*), and 3) the genetic background (BGND—JU1200 or JU751). To do this, we generated a large data set containing all eight possible genotype combinations (labelled i to viii) (Fig. 6a). Analysing this data set revealed both subtle and dramatic two- and three-way interactions (Fig. 6b–e). To explore these data in detail, we built a generalized linear model, using a top-down model selection strategy to optimize the Bayesian information criterion (BIC) and the adjusted deviance accounted for in the model ($D^2$)[78] (Supplementary Note 11). The optimized model indicated that PZ size is best explained by all three main effects and all their interactions (i.e., fully crossed). In addition, the model includes a fixed main block effect and interactions to account for variation across different experimental blocks. The model accounts for 33.4% of the adjusted deviance ($D^2$) and fits the data well (residual deviance = 2796.5 on 2763 df; $P\chi^2$ = 0.324) (Supplementary Note 11).

To interpret the model, it is helpful to compare the estimated marginal means (Fig. 6b, Supplementary Note 11b). Examining genetic interactions of the two loci and genetic background reveals widespread and strong epistasis explaining variation in PZ size. As mentioned earlier (Fig. 5b), the *lag-2(cgb1007)* deletion has opposing phenotypic effects depending on the genetic background: the deletion strongly reduces PZ size in JU1200 but increases PZ size in JU751 (Fig. 6b). Examining the epistatic interactions shows that the effects of

the *lag-2(cgb1007)* variant are highly contingent on both C2 and genetic background, so that the *lag-2* deletion may have negative, positive or no phenotypic consequences across the set of genotypes (Fig. 6b–e). Similarly, the phenotypic effects of C2 are strongly dependent on the genetic context: C2-JU751 reduces PZ size in every genotype (Fig. 6b–i vs. ii, v vs. vii, and vi vs. viii) *except JU1200<sub>C5-lag-2p(-)</sub>* (Fig. 6b–iii vs. iv). The presence of the *lag-2* deletion dampens the effect of *C2-JU751* in the JU1200 background (Fig. 6e' and e''), which could imply that alleles within C2 act, at least in part, through this region of the *lag-2* promoter. As one might expect, the combined effect of all other unidentified variants in the genetic background has the greatest influence on PZ size overall. Yet, this effect does vary remarkably depending on the presence of *lag-2p* deletion and genotypes at C2. In general, the JU751 background strongly reduces PZ size (Fig. 6c' and c''), but there is no significant difference between *JU1200<sub>C2-JU1200</sub>* and *JU751<sub>C2-JU1200</sub>* when the *lag-2* deletion is present (Fig. 6b–iv vs. v, Fig. 6c'). We interpret this to mean that, in presence of the *lag-2* deletion (*lag-2p(-)*), *C2-JU1200* variants exert a strong positive effect to maintain PZ size despite the average negative effect of variants in the JU751 background (Fig. 6d'', red line). Conversely, *C2-JU751* variants cannot maintain this positive effect resulting in background sensitivity (Fig. 6d'', blue line). Finally, without the *lag-2p* deletion (*lag-2p(+)*), the *C2-JU751* variants do not modify the strong negative effect of the JU751 background (Fig. 6d' vs. Fig. 6d'' red lines).

Our model shows that higher-order epistatic interactions contribute to variation in *C. elegans* PZ size. Importantly, some of the

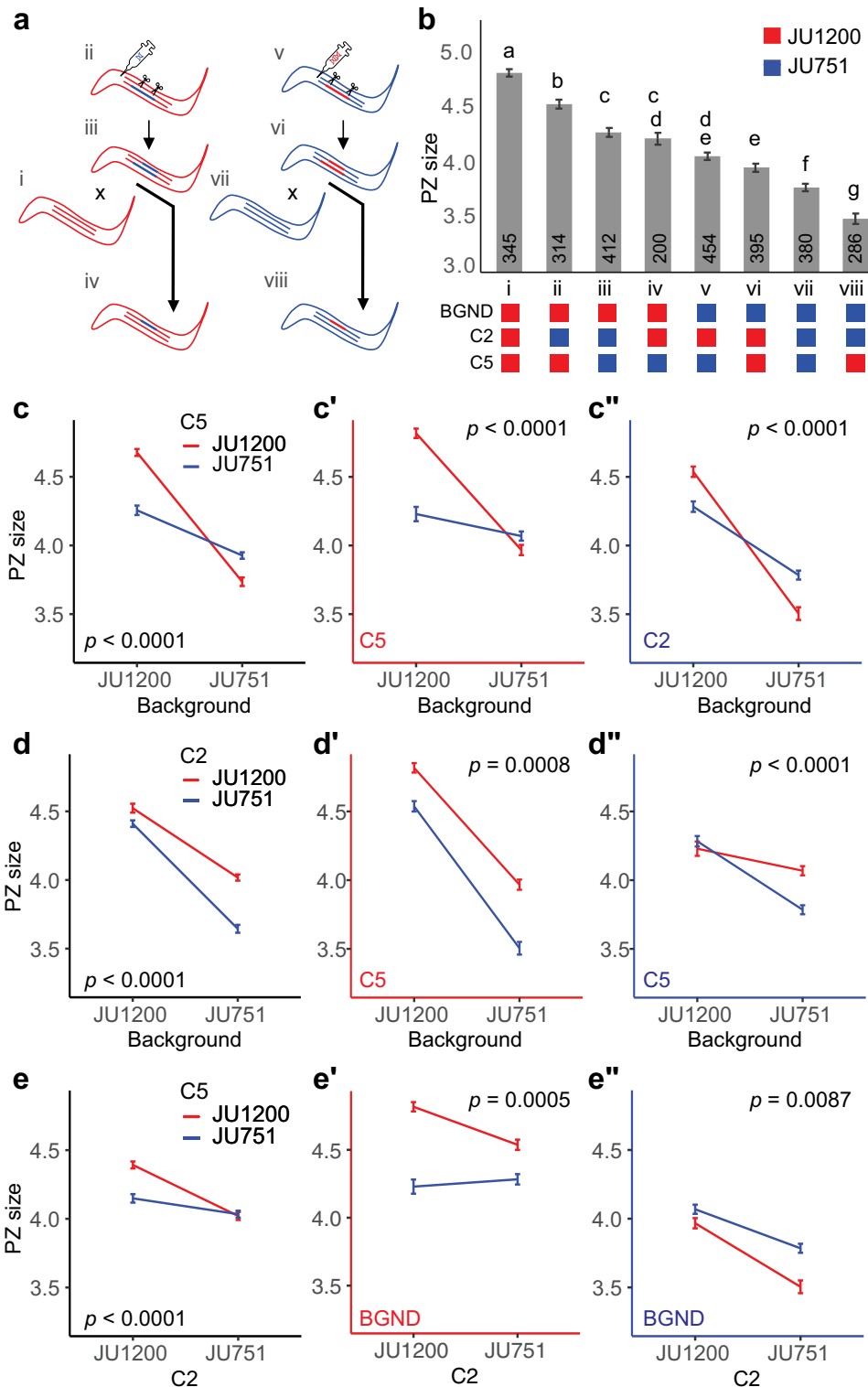

observed phenotypic effects are masked when only two-way interactions are considered (for example, Fig. 6e vs. e' and e"). These results highlight the need for caution when interpreting the phenotypic effect of a variant or allele, as the effect may be highly contingent on other variants in the genetic background. We also stress that even our interpretations of the effects of C2 and the genetic background must be considered with care as they are aggregate effects, which may be influenced by underlying epistatic interactions and, possibly, additive effects of closely linked loci.

## Discussion

Our study explores the natural variation and quantitative genetic architecture of a germ stem cell system. Examining only a handful of *C. elegans* wild isolates, we were able to detect significant differences in germline progenitor zone (PZ) size, indicative of variation in germ stem cell niche activity. Our main findings are: (1) Natural variation in PZ between two isolates maps to multiple, partly interacting QTL. (2) A QTL on chromosome V contains a deletion, unique to JU751, in the promoter region of the DSL ligand *lag-2*, a known key signal promoting

**Fig. 6 | Higher-order epistatic interactions shape natural variation in PZ size.**
**a** Scheme depicting the generation of the eight genotypes used in the interactions analysis. The different genotypes were derived by made either creating or replacing the *lag-2(cgb1007)* deletion in JU1200 and JU751 background NILs. The successfully modified lines were then backcrossed to the parental lines to isolate the CRISPR modifications in the parental backgrounds. Genotypes are labelled i-viii for clarity. **b** Estimated marginal means ± standard errors of PZ size from a generalized linear model describing a data set containing all eight genotypes. JU1200 and JU751 genotypes are indicated by red and blue, respectively, and are indicated for each of the loci and background below the chart. Crossbars and error bars represent estimated marginal means ± standard errors from a generalized linear model (two-

sided). Lowercase letters indicate significant ($p < 0.05$) Tukey-adjusted pairwise contrasts such that groups that share a letter are not significantly different. *n*-values across six blocks for each genotype are indicated in the bars. **c–e** The same model-derived means ± standard errors as in panel **b** presented to show interactions among the loci and background. Graphs with black axes show only two-way interactions. The third dimension is represented as a split into two graphs with red and blue axes indicating the genotypes for the locus given in the bottom left. Two-way interaction *p*-values are given in the graphs. Analyses were carried out on raw data (PZ area in pixels), but the y-axes were scaled to µm² for presentation (0.0504 µm²/pixel) (see Supplementary Note 11 for model details). Source data are provided as a Source Data file.

the germ stem cell fate via the Notch pathway. (3) Natural quantitative variation in *C. elegans* germline proliferative activity arises through modulation of transcriptional activity of core signalling elements of the germ stem cell niche. (4) Introgression and allelic replacement lines reveal that the main effects and interactions of the two QTL on chromosomes II and V (*lag-2* deletion) are partly antagonistic and highly contingent on the genetic background. (5) Hence, higher-order epistatic interactions shape natural variation in germ stem cell niche activity.

Our observations suggest that *C. elegans* germ stem cell niche activity is a typical complex trait involving a polygenic architecture. Additional observations support this view indirectly: (1) PZ size varies continuously, and often subtly, in natural populations (this study); (2) the *lag-2* deletion was not found in other wild isolates with small PZ size (this study); and (3) mutations in a large number of genes modulate PZ size and these genes exhibit diverse functions that not only act in germ stem cell proliferation, cell cycle progression, and differentiation, but also in diverse metabolic or sensory processes (reviewed in[25,39]). Together with our findings, these observations suggest that *C. elegans* PZ size is a higher-order phenotype that likely integrates effects of allelic variation at many loci. In the case of our example, focusing on two target isolates with pronounced differences in PZ size, we also identified large-effect loci or variants on chromosomes II and V whose effects were partly antagonistic and, overall, strongly dependent on the genetic context. Together, and in line with previous studies, this suggests that the effects of large-effects variants, even if resolved at the molecular level (such as the *lag-2(cgb1007)* deletion), can be misleading if epistatic interactions in the native background are ignored[4,79]. Our work shows that generating allelic replacements in NIL backgrounds is an effective means of uncovering these interactions, even when the specific variants are not known.

Performing linkage QTL mapping on a panel of F2 RILs, derived from two parental isolates with divergent PZ size, we detected four, partly interacting QTL. In the best model identified by our multi-QTL model selection approach, the QII and QV QTL act additively to explain 32% of the phenotypic variance in the examined RIL population. However, analysis of introgression and allelic replacement lines uncovered extensive epistatic interactions among these loci and the genetic background. This finding is not surprising given that gene-gene interactions contribute to the component of additive genetic variance ($V_A$) measured at the population level; that is, high $V_A$ variance need not reflect additive gene action[7,80–87]. In particular, measures of $V_A$ due to individual loci will strongly depend on allele frequencies of all loci that are epistatic to each other. Hence, measures of $V_A$ (and other genetic variance components, including the epistatic variance) provide little to no information about underlying epistatic interactions, and hence, genetic trait architecture, even when QTLs have been detected. Characterizing the molecular nature of QTLs and their interaction in controlled genetic backgrounds is, therefore, essential.

Resolving the QV QTL region identified a unique 148 bp deletion in the *lag-2* promoter region of JU751, *lag-2(cgb1007)*. This deletion removes one of six E-box sites required for binding of HLH-2, a positive regulator of *lag-2* mediated germ cell proliferation and germline

expansion[30,73]. Mutating these E-box sites was shown to result in reduced *lag-2::GFP* transgene expression specifically during the L3 stage (and subsequently reduced germ cell proliferation)[73]. We, therefore, expected *lag-2(cgb1007)* to cause reduced *lag-2* expression due to reduced HLH-2 binding activity. Consistent with this scenario, introducing this deletion into JU1200 (with a large PZ size) significantly reduced both PZ size and *lag-2* expression in the DTC (Fig. 5b, e). In contrast, restoring the corresponding ancestral sequence (JU1200) of this deletion in JU751 (with a small PZ size) further reduced, rather than increasing, PZ size along with *lag-2* expression (Fig. 5b, f). The *lag-2* deletion allele on its own, therefore, exerts opposing effects depending on the genetic background (sign epistasis). However, examining the interaction between the *lag-2(cgb1007)* deletion (C5) and a central portion of the chromosome II QTL (C2) shows that *C2-JU1200* can dampen this reversal (Fig. 6c' vs. c"). In addition, the *lag-2* deletion modifies the effect of C2 by dampening the negative effects of *C2-JU751* in both backgrounds (Fig. 6e' vs. e"). This indicates that C2 interacts with the *lag-2* promoter region, specifically in the region of the *lag-2(cgb1007)* deletion. We do not have any hypotheses regarding the molecular nature of this interaction as none of the genes (*hlh-2, lin-39, unc-130, daf-3/daf-5, ces-1*) with confirmed binding sites or response elements in the *lag-2* promoter region are located within the C2 region. Moreover, although coding polymorphisms (JU1200 versus JU751) exist in some of these genes (*lin-39, unc-130,* and *daf-5*) none occur in *hlh-2, daf-3* and *ces-1*[70]. Resolving the QII QTL would, therefore, be required to dissect this genetic interaction. Overall, the *lag-2* deletion containing an HLH-2 binding site previously shown to positively effect *lag-2* expression in N2, can have positive, negative, or neutral effects depending on the genetic context (Fig. 6).

Although it remains unclear whether and how the *lag-2(cgb1007)* deletion contributes to reduced PZ size in JU751, our smFISH measurements of *lag-2* transcripts in the DTC provide unambiguous support that differences in *lag-2* transcriptional activity contribute to differences in germ stem cell niche activity observed between JU751 and JU1200. *lag-2* smFISH measurements closely mirrored PZ size measurements not only in parental isolates but also in reciprocal ARLs, in which the *lag-2* promoter fragment was manipulated (Fig. 5b). Identification and characterization of this variant show that natural variation in PZ size can occur through direct modification of core signals acting in the *C. elegans* stem cell niche. However, we cannot rule out additional contributions from other unidentified variants in the QV QTL. Likewise, while we did not observe any obvious effects on other larval developmental processes involving Notch signalling (including regulation mediated by HLH-2) in strains carrying *lag-2(cgb1007)*, such as the AC/VU cell fate specification[30,73], we did not specifically analyse them and, therefore, cannot rule them out.

While the activity of any germ stem cell system is obviously relevant for organismal reproductive fitness, it is not clear whether and how observed natural variation in germ cell proliferation (and PZ size) might translate into variation in reproductive fitness. Although increased germ cell proliferation has been found to correlate with increased egg-laying activity, offspring production, and egg quality[10,25,31,32,58,88,89], it is unclear to what degree this relationship is

causal: *C. elegans* primarily reproduces through self-fertilizing hermaphrodites, which sequentially produce sperm, then oocytes for the remainder of life. In laboratory conditions, this causes *C. elegans* fecundity to be limited by the amount of self-sperm (~250) initially produced. PZ size is generally measured, as in our study, during the early adult stage, so that likely many of the observed progenitor cells will never develop into mature oocytes and become fertilized under selfing. However, increased germ cell proliferation allows for improved oocyte quality by upregulating the flux and number of oocytes undergoing physiological cell death (apoptosis), hence liberating resources to provision surviving oocytes[10,25,31,32,58,88,89]. Therefore, while adult PZ size may not be a direct proxy for future reproductive potential (offspring number), it may enhance offspring quality through improved oocyte provisioning. In addition, adult PZ size also reflects the past proliferative activity of earlier larval stages, as illustrated by our quantifications of proliferative activity (Figs. 2b, 5d). Variation in adult PZ size could thus reflect differential reproductive investment during larval development, which may trade-off with energy allocation to somatic development.

Regarding reproductive fitness differences of the target isolates in our study, JU751 and JU1200, we have previously shown that selfing JU751 hermaphrodites have significantly reduced brood size relative to JU1200[71]. This reduction is not caused by differential sperm production but partly due to a major-effect variant causing early matricidal hatching in JU751[71]. Still, even after correcting for this genetic variant in JU751, brood size remains significantly smaller in JU751 relative to JU1200[71]. It is possible that reduced germ cell proliferation (and smaller adult PZ) in JU751 reflects a lower reproductive investment. Of course, this scenario is highly speculative and observed differences need not be adaptive. Similarly, we ignore whether any of the detected QTL, including the *lag-2(cgb1007)* deletion, are maintained by selection. Even if selectively advantageous it remains unclear whether QTL were selected because of their effects on germline proliferative activity or because of potential pleiotropic effects on additional processes outside the germline, known to involve Notch signalling during larval development[30,73].

Here we focused on the analysis of two large-effect QTLs and their interactions, detected using a rather small panel of RILs (n = 70). Therefore, detecting small-effect and additional epistatic interactions through linkage mapping were unfeasible given the relatively low statistical power. Nevertheless, a specific search for interacting loci with opposing effects yielded two candidate QTL on chromosomes I and X (Fig. 3h). Further analysis of this antagonistic interaction will likely be limited by its relatively small effect size and the already complex interactions between the chromosome II QTL, *lag-2(cgb1007)*, and the genetic background. In addition, further fine-mapping experiments are required to fully understand epistatic interactions between the QII and QV QTL uncovered here. Both QTL could harbour multiple loci affecting PZ size variation, potentially also including closely linked, antagonistically loci, which seem to be common in *C. elegans*[79].

Beyond these technical limitations, several other issues complicate the interpretation of our findings. First, and foremost, artificial mapping populations, such as the F2 RIL panel used here, may generate novel epistatic interactions through disruption of linked genomic regions. This is particularly relevant for *C. elegans*, a predominantly selfing species, with low effective recombination leading to strong linkage disequilibrium[74,90–92]. Most prominently, crosses between wild isolates of selfing *Caenorhabditis* have uncovered consistent, strong outbreeding depression due to widespread genetic incompatibilities reducing survival and reproduction[91,93–96]. Such epistatic interactions of novel allelic combinations generated in artificial *C. elegans* populations are thus particularly likely to affect traits that involve a polygenic architecture[4,17,79,97,98]. In other words, epistatic interactions observed in artificial mapping panels need not reflect epistatic interactions that occur in the wild. Nevertheless, their

study does provide insight into the genetic architecture underlying trait variation.

Together with past research, our study reinforces the view that generalizations on trait architecture based on the isolated study of individual molecular variants in single genetic backgrounds are likely misleading. Combining quantitative and developmental genetic approaches is, therefore, essential to understanding complex quantitative trait architecture in the face of epistatic and often idiosyncratic interactions. Given that currently (and likely for a long time) even the most sophisticated experiments can interrogate only a tiny fraction of all possible epistatic interactions, integrating developmental and quantitative genetic approaches affords the best option to bring us one step closer to understanding the genetic basis of phenotypes and their variation.

## Methods
### Experimental model
All *C. elegans* strains used in this study are listed in Supplementary Data 2. Standard *C. elegans* methods were used to maintain all strains at 20 °C on 2.5% agar NGM plates seeded with *E. coli* strain OP50[99–101]. All experiments were performed on hermaphrodites. All strains or biological materials are available from the corresponding authors upon request.

### Quantification of germline progenitor zone nuclei
Images of dissected gonads fixed in 100% ice-cold methanol and stained with DAPI (VECTASHIELD® Antifade Mounting Medium with DAPI) were taken using an Olympus BX61 microscope with a CoolSnap HQ2 camera. Z-stacks with 1 μm increments were performed in the DAPI channel at a 40x magnification. The nuclei were counted by hand using ImageJ2 v2.9.0/1.53t[102]. First, the PZ was defined as the region adjacent to the distal tip cell ending at the transition zone boundary where two or more crescent shape nuclei per row of germ cells can be observed[38]. The stack of images was cropped at the progenitor zone to the transition zone boundary and nuclei were counted.

### Quantification of germ cell mitosis
Mitotic germ cell proliferation was quantified in L4 larvae using the Click-IT EdU kit from Thermofisher (Cat: C10338) according to the manufacturer's instructions unless otherwise noted. Briefly, worms were synchronized via hypochlorite treatment (1:1:2 Bleach:1 M NaOH:dH$_2$O for 6 min then spun and washed 3x in M9 buffer) and eggs were allowed to hatch overnight in M9 buffer at 20 °C. Eggs were plated at a density of ~400 eggs/plate and allowed to develop until L4. Mid-L4 larvae or young adults (1-10 eggs *in utero*) were isolated, washed, and allowed to soak in 100 μL of 20 mM EdU in M9 buffer for exactly 15 minutes (sufficient time to stain 50-75% of the PZ in young adults). Worms were quickly washed in M9 to remove the EdU, and the gonads were released by cutting on adherent glass slides (Fisherbrand™ Superfrost™ Plus Microscope Slides). A small amount of levamisole was used to anesthetize the worms. Dissected gonads were carefully washed on the slides, fixed in 4% PFA, and washed again. Washes were performed by application of small volumes of solution directly to the tissue on the slides followed by careful aspiration. After a final wash in dH$_2$O, tissue was dried on a slide warmer (35 °C for 5 min) to adhere tissue to slides. Slides were soaked in 100% MeOH at −20 °C overnight. The following day, tissues were rehydrated in PBST and EdU staining was performed using the Click-IT EdU kit according to the manufacturer's instructions. The Click-IT staining protocol was repeated once and then slides were mounted with VECTASHIELD® Antifade Mounting Medium with DAPI. Imaging was performed through a 40x objective on an Olympus BX61 microscope with a CoolSnap HQ2 camera. Z-stacks were taken though the distal gonad (one per worm). EdU-positive and DAPI-stained PZ nuclei were counted by hand using ImageJ2 v2.9.0/1.53t[102].

## Estimation of PZ size

Quantifying PZ germ cell number can be a rate-limiting step in the scoring of PZ size among many samples. To overcome this obstacle, we developed a simple method of estimating PZ size and used this as a proxy for PZ cell number (Fig. 1b and c). In brief, lines were synchronized by hypochlorite treatment (1:1:2 Bleach:1 M NaOH:dH$_2$O for 6 min then spun and washed 3x in M9). Eggs were allowed to hatch in M9 buffer at 20 °C overnight. The following day, the eggs were plated at a density of ~400 worms/plate on standard NGM media seeded with OP50. Larvae were allowed to develop into young adults at 20 °C. Young adults were collected by washing the plates when most animals showed 1-10 eggs *in utero*. Samples were washed in M9 and fixed in ice-cold methanol for at least 5 min. To prepare samples for imaging, worms were rehydrated in M9 and applied to glass slides with Vectashield mounting medium containing DAPI (Vector Laboratories, Burlingame, CA). The distal germline regions of young adult hermaphrodites (containing 1–10 embryos *in utero*) were imaged through a 40x objective on an Olympus BX61 microscope with a CoolSnap HQ2 camera. One gonad arm was imaged per worm, that being the arm which happened to be closer to the objective lens. A single image was taken for each worm, with the plane of section capturing most of the nuclei on one side of the gonad such that crescent-shaped nuclei were visible, and the shape of the gonad in that plane was representative of most planes (Fig. 1b). PZ size was then quantified for each image. The progenitor zone was defined as the region adjacent to the distal tip cell ending at the transition zone boundary where two or more crescent-shaped nuclei per row of germ cells can be observed[38]. The PZ was cropped away from all other tissues using ImageJ2 v2.9.0/1.53t[102], and the threshold tool was used to maximize the number of PZ nuclei highlighted while minimizing any highlighted background. The number of pixels highlighted was recorded as PZ Area. This method of estimation correlates well with hand counts of PZ nuclei (Fig. 1c).

## RIL phenotyping and linkage QTL mapping

We quantified PZ size in a set of 70 SNP-genotyped RILs derived from the two parents, JU1200 and JU751[71,103]. Lines were divided into six scoring blocks and scored over the course of eight weeks. While some strains were measured in two separate blocks, not all were. Most strains were measured only once, with 30–40 individuals per strain. The averages and variances for strains that were measured twice were similar between measurements. Only one set of measurements was used for the QTL mapping. PZ size was normalized across blocks using a correction factor. The correction factor was derived from the least square means of each block (lsmeans package v. 2.30-0) to each observation (correction factor = LSM$_{BlockX}$/LSM$_{Block1}$). Normalized PZ size was mapped using the software package R/qtl (v. 1.50)[72]. First, a genetic linkage map was derived from the SNP genotypes of all 70 RILs. Then, single QTL and two-QTL standard interval mapping based on hidden Markov models was used to identify candidate QTL. LOD thresholds were set to the 95$^{th}$ percentile of the maximum genome-wide LOD scores derived from 5,000 random permutations of the data under the global null hypothesis of zero QTL. These candidate QTL served as the starting point for multi-QTL modelling in which the R/qtl software generates penalized LOD scores for each tested model. Models that perform better than the null hypothesis of zero QTL have penalized LOD > 0. We selected the model with the highest penalized LOD score as recommended by the authors of R/qtl (Fig. 3j)[72].

## NIL generation and chromosome II QTL genotyping

To validate the genomic interval of the QTL on chromosome II, we constructed nearly isogenic lines (NILs). Two RILs with JU1200 sequence in the QTL region (RIL128/NIC728 and RIL71/NIC671) and two RILs with JU751 sequence in the QTL region (RIL127/NIC727 and RIL54/NIC654) were backcrossed for 10–12 generations to isolate

the region in the opposite parent's background. Lines were selected at each generation based on PCR genotyping of several INDELS in the chromosome II QTL (see Supplementary Data 1 for primers and Supplementary Data 2 for lines created and their genotypes).

## Generation of allelic replacement lines (ARLs) and CRISPR/Cas-9 gene editing

To validate the candidate INDEL upstream of *lag-2* in the chromosome V QTL, we created reciprocal allelic replacement lines (ARLs) using CRISPR/Cas-9 editing using a protocol similar to that described by[104]. To introduce the deletion (*cgb1007*) into the JU1200 background, we selected a NIL containing JU751 sequence through most of the chromosome II QTL (NIC1701) and injected sgRNAs (Synthego Corp.) to induce a double cut around the insertion site in the *lag-2* promoter (Supplementary Fig. 2, Supplementary Data 1). We co-injected a single-stranded donor oligonucleotide to act as the repair template (Supplementary Data 1). *dpy-10* was used in a co-CRISPR strategy[104]. Lines from separate injections containing successful replacements in *lag-2p* were identified by PCR genotyping of the deletion and Sanger sequencing. These lines were then crossed to JU1200 to segregate away the chromosome II containing JU751 sequence to generate lines containing both the JU751 version of the chromosome II QTL and the *lag-2p(cgb1007)* deletion (NIC1713, NIC1714, NIC1715) as well as a line containing only the *lag-2p(cgb1007)* deletion in the JU1200 background (NIC1720). To replace the ancestral *lag-2p* deletion sequence (*cgb1008*) in the JU751 background, we selected a NIL containing JU1200 sequence in the chromosome II QTL (NIC1672) and injected guide RNAs (Synthego Corp.) (Supplementary Data 1) to induce a double cut around the insertion site. We co-injected a double-stranded PCR product containing the ancestral (JU1200) sequence with *lag-2p(cgb1008)* to act as the repair template (Supplementary Data 1). *dpy-10* was used in a co-CRISPR strategy according to[104]. Lines from separate injections containing successful replacements were identified by PCR genotyping of *lag-2p(cgb1008)* and Sanger sequencing. These lines were then crossed to JU751 to segregate away the chromosome II containing JU1200 sequence, leaving us with lines containing both the JU1200 version of the chromosome II QTL and *lag-2p(cgb1008)* (NIC1716, NIC1717, NIC1719) as well as lines containing only *lag-2p(cgb1007)* in the JU751 background (NIC1723, NIC1724, NIC1725). For a table of all lines created and genotypes, see Supplementary Data 2.

## Measurement of lag-2 transcripts via smFISH

Worms were bleach-synchronized as above and grown at 20 °C until the appropriate developmental stage. Worms were then washed off plates and fixed in 1 mL fresh fixative (4% formaldehyde in PBS) for 40 min. Worms were washed twice in PBS, resuspended in 1 mL 70% ethanol, and stored at 4 °C for several days. A *lag-2* RNA probe set (Barkoulas et al. 2013) coupled to AF594 (Custom Stellaris Fish Probes, Biosearch Tech, Teddington UK) was resuspended in RNase-free TE buffer (pH 8) to make a 100 μM stock then diluted 1:30 (working solution) in RNase-free water. Fixed worms were washed 3 min in 1 mL wash solution (10% deionized formamide and 2x Saline Sodium Citrate buffer (Ambion) in RNase-free water. Worms were then resuspended in 100 μL hybridization solution (10% formamide, 10% dextran sulfate, 2x SSC) + 1 μL of the *lag-2* probe working solution and incubated in the dark at 30 °C overnight. The following day, samples were rinsed, washed 30 min at 30 °C in wash solution, and stained 30 min at 30 °C in 1 mL wash solution + DAPI (7.5 μg/mL final, Sigma). Finally, samples were washed twice in PBS and mounted on glass slides in Prolong diamond antifade mounting medium (Eugene, OR). Worms were imaged under epifluorescence on an inverted Zeiss Z.1 microscope with a 100x oil immersion objective. The same illumination and exposure settings (Zen 3.2 Blue Edition software) were used for all samples. Z-stacks were taken through the DTC with a step size of

0.4 µm and fluorescent puncta were quantified by hand using ImageJ2 software (NIH, Bethesda, MD, http://rsb.info.nih.gov/ij/)[102].

## Quantification and statistical analysis

Statistical analyses were performed using R Statistical Software (R version 4.2.0, RStudio 2022.02.3 build 492) using generalized linear mixed models[105]. Data were fitted to either gaussian or negative binomial (nbinom2) models using the glmmTMB package v. 1.1.3[106]. Residual diagnostics were performed using the DHARMa package v. 0.4.5 according to the developer's guidelines[107]. When model selection was warranted, the Bayesian information criterion (BIC) was used to select the optimal model. The emmeans package v. 1.7.4-1 was used to calculate model estimated marginal means, standard errors, and Tukey-corrected p-values for pairwise contrasts[108]. For model fitting of the interaction data set corresponding to Fig. 6, $D^2$ was calculated using the modEvA package v. 3.0[78]. Specific details of each data analysis are described in the Supplementary Note legends, which display the statistical results (Supplementary Notes 1 to 12). Other R packages used for data manipulation and plotting included tidyverse v. 1.3.1[109], MASS v. 7.3-57[110], ggbeeswarm v. 0.6.0[111], rnaturalearth and rnaturalearthdata v. 0.1.0[112], rgeos v. 0.5-9[113], and sf v. 1.0-7[114].

## Reporting summary

Further information on research design is available in the Nature Portfolio Reporting Summary linked to this article.

## Data availability

All raw data are provided in the Source Data file. The following publicly available databases were referenced and used in this study: Wormbase WS283 (https://wormbase.org/#012-34-5) and the *C. elegans* Natural Diversity Resource (https://elegansvariation.org/). This paper does not report any original algorithms. Source data are provided with this paper.

## Code availability

Only limited code was generated to use the R packages for statistical analysis and to generate custom plots according to the developers' guidelines (See Supplementary Notes 1-12 and the Methods).

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

## Acknowledgements

We thank Nausicaa Poullet, Anne Vielle, and Clotilde Gimond for contributions to experimental work that initiated this project. We thank

Fabien Duveau, Marie-Anne Félix, Luke Noble, Alistair McGregor, Clotilde Gimond, Laure Mignerot and Joao Picao Osorio for discussion, helpful comments on previous versions of the manuscript, and technical advice. *C. elegans* strains were kindly provided by the *Caenorhabditis elegans* Natural Diversity Resource (*Ce*NDR) and Marie-Anne Félix. We would also like to thank *Ce*NDR (https://elegansvariation.org/) and WormBase (https://wormbase.org/) for providing resources without which the analyses performed here would not have been possible. This study was supported by project grants from the Fondation ARC sur la recherche sur le cancer (PJA 20161205047 to C.B.) and the Agence Nationale de la Recherche (ANR-17-CE02-0017 to C.B.) SRF was supported by a post-doctoral fellowship from the City of Nice, France (Ville de Nice: Aides Individuelles Jeunes Chercheurs). We acknowledge additional institutional support by the Centre National de la Recherche Scientifique (CNRS), the Institut national de la santé et de la recherche médicale (Inserm), and the Université Côte d'Azur (UCA).

## Author contributions

Conceptualization, S.R.F. and C.B.; Methodology, S.R.F., A.S., C.B.; Investigation, S.R.F., B.B., and A.S.; Formal Analysis and Visualization, S.R.F.; Writing—Original Draft, S.R.F.; Writing—Review and Editing, S.R.F., C.B.; Supervision and Project Administration, C.B.; Funding Acquisition, C.B., S.R.F.

## Competing interests

The authors declare no competing interests.
