## [Peer Review File · Nature Communications]

Higher-order epistasis shapes natural variation in germ stem cell niche activityREVIEWERS' COMMENTS

Reviewer #1 (Remarks to the Author):

In this manuscript, Fausett and colleagues pick apart the genetic mechanisms of natural variation in the *C. elegans* stem cell niche. They observed that several wild strains have different sizes of proliferation zones and then turned to a linkage mapping approach between two isolates. They detect several QTL (both additive and interacting) involved in this trait, and perform crosses and allele replacements to validate (two large-effect QTL) and causally prove variation in one of the two QTL. Then, they go on to create a series of strains to test background effects and interactions between QTL detected in the mapping approach. They find convincing evidence of higher order epistatic interactions. The manuscript is well written and the data are clear. I only have a few minor suggestions.

(1) The use of "causal" in the Summary is appropriate for the QTL on V but not for the QTL on II (where no variant was causally connected to the progenitor zone (PZ) variation. Please reword.

(2) It is perhaps stylistic, but I prefer linkage mapping over QTL mapping. Both linkage and association mapping detect QTL. It would be good to edit throughout the text.

(3) Figure 1C. How many different strains with different individuals go into these data? Please describe more in the Methods and alter the figure to note the replication.

(4) The RIL panel needs more description. In the supplement, the authors should provide the map object from R/qtl and the figures assessing expansion. It would be good for readers to be able to verify that the small number of markers reflect an even genetic map.

(5) Figure 3C. What is the significance line? Presumably GWER, but it would be good to specify in the Figure and Methods.

(6) What is the broad-sense heritability for this trait? Also, why type of replication did the authors perform for the linkage mapping experiment? Please add these details to the Methods and Figure descriptions.

(7) Figure 3D and G. Why not show the two-component QTL scan for the entire genome? I'm interested to see how the other chromosomes look.

(8) Figure 4A and B. What are the vertical ticks for each strain genotype?

(9) Figure 5B. I think it would be more clear if the genetic backgrounds were plotted next to each other (e.g., JU200, JU1200 edit, JU751, JU751 edit).

(10) The authors do a great job explaining the importance of their results and how epistasis can impact quantitative variation in the Discussion. In the second paragraph, it might be good to suggest that investigators can use the NIL backgrounds for allele replacements to avoid uncharacterized epistatic interactions from the genetic background. They have good evidence that this approach would help.

(11) The authors should cite WormBase.

Reviewer #2 (Remarks to the Author):

This manuscript examines the surprisingly complex genetic architecture that governs quantitative

variation in the size of the germline mitotic zone (or proliferative zone, PZ) in *C. elegans*. That variation between genotypes can be reproducibly detected is itself somewhat surprising, as it is presumably under strong stabilizing selection in all populations, and also famously sensitive to environmental influences that make measurements challenging. Nevertheless, the authors have convincingly demonstrated that inbred isolates do vary reproducibly, and that recombinant inbred lines (RILs) can be used to identify specific loci that influence PZ size. This is an elegant and very impressive study that focuses on one such locus, a polymorphic deletion in the promoter of *lag-2*, a gene with a long history of association with PZ function that encodes the Notch ligand LAG-2.

The authors use cutting-edge methods to identify the *lag-2* QTL, and to explore its complex epistatic relationships with other variable sites. They are meticulous in their genetic and statistical approaches, and I really have no technical concerns. The manuscript is also appropriately cautious about the results. So, overall this is an impressive study.

My one substantial concern is that the paper quickly dives into the minutiae of measuring PZ variation and its dependence upon genotype, without explaining whether there is any organismal consequence of the variation. For example, does PZ size correlate with total brood size, or egg laying rate? If so, there may be an adaptive angle on all of this. If not, we may instead be seeing "allowable slop" that is tolerated because the overall reproductive phenotype is not tightly connected to the PZ dynamics described here. Some sense of whether any of this ultimately matters to the worms would help motivate digestion of the many genetic details.

Beyond the above major issue, there are two smaller points in the manuscript that could be clarified:

Summary, line 11: This was initially unclear to me. After reading the paper, I think an improved wording would be "Unexpectedly, restoring the deleted ancestral sequence in the isolate with a smaller PZ did not increase its PZ, but instead further reduced PZ size."

Page 4, section "Germ cell proliferative activity differs between wild isolates JU1200 and JU751": Looking at Figure 1, one immediately wonders why the authors did not use the more extreme pairs, like JU1200 vs. CB4856, in their study. My guess is that it was because RILs already existed for these two strains. That is a good reason, but stating it explicitly here would help the reader better understand the choices made.

Reviewer #3 (Remarks to the Author):

Mechanisms impacting quantitative developmental traits are of great interest for system-level analysis, with the goal of reconstructing the regulatory network determining specific outcomes. One of those outcomes is the number of stem and progenitor cells maintained in the niche that regulates stem cell number on a population level. The manuscript by Faucett and colleagues identifies a major weakness of the current approach to these studies that does not consider natural variation. The focus of this manuscript is the regulation of germline progenitor zone (PZ) size in *C. elegans*, focusing on two natural isolates that maintain distinct PZ sizes. Through quantitative trait locus analysis, the study identifies candidate loci responsible for distinct PZ sizes. One of these is a deletion in the promoter of the Notch signal essential for PZ maintenance found in a strain with the short PZ, and sufficient to decrease PZ size in the other genetic background. However, restoring the ancestral sequence of the *lag-2* promoter further reduced PZ size, suggesting higher-order epistasis with additional genomic loci. Therefore, focusing the studies of regulatory network impacting PZ size on the single genetic background will miss important regulatory interactions that only come to light after considering natural variation.

Previous evolutionary analysis of transcriptional regulatory regions already appreciated significant

heterogeneity of regulatory modules in natural populations (eg, Balhoff and Wray, 2005; PMID: 15937122). Importantly, the study by Faucett et al was able to use gene editing to rigorously investigate the contribution of a specific promoter deletion to a quantitative trait, and distinguish the effect of this single locus from the effects of the general genetic background as well as from the effect of broader QTL loci. Therefore, it constitutes a significant advance in the field.

This is a thorough and well-written manuscript, that would benefit from the following suggested modifications:

1. To enhance data transparency, I advise providing the N numbers for PZ size analysis in all figures or explicitly referring to the supplementary dataset that contains the raw data. A good place for these would be Supplemental Tables already summarizing much statistical data.

2. Since *lag-2* is a signal involved in many developmental fates in addition to germ line stem and progenitor cell maintenance, it would be helpful if the authors commented on any extra-germline phenotypes consistent with a reduction in Notch signaling during larval development (for example, disruption of AC/VU cell fate specification, which is also thought to be dependent on HLH-2-mediated regulation). This would help establish whether the effects of the deletion are tissue-specific or affect multiple developmental events.

Response to Reviewers

We would like to thank all the reviewers for their thoughtful comments, criticisms, and suggestions. We have revised our manuscript and figures to incorporate all suggestions. Please find below our responses (in blue) to each individual reviewer comment.

Reviewer #1 (Remarks to the Author):

1) The use of “causal” in the Summary is appropriate for the QTL on V but not for the QTL on II (where no variant was causally connected to the progenitor zone (PZ) variation. Please reword.

Response: The Summary has been substantially revised to comply with the 150-word limit. The term ‘causal’ is no longer used.

(2) It is perhaps stylistic, but I prefer linkage mapping over QTL mapping. Both linkage and association mapping detect QTL. It would be good to edit throughout the text.

Response: Throughout the manuscript, we have changed the term “QTL mapping” to ‘linkage’ or “linkage QTL” mapping.

(3) Figure 1C. How many different strains with different individuals go into these data? Please describe more in the Methods and alter the figure to note the replication.

Response: Additional detail has been added to the Fig. 1c legend to clarify. The complete description of the data and analysis is provided in the Supplemental Tables. In addition, we have added sample size information for all other figures in the manuscript.

(4) The RIL panel needs more description. In the supplement, the authors should provide the map object from R/qtl and the figures assessing expansion. It would be good for readers to be able to verify that the small number of markers reflect an even genetic map.

Response: As mentioned in the manuscript, this RIL panel was established in a previous study from our group (Vigne et al. 2021. *Science Advances* 7 (6). <https://doi.org/10.1126/sciadv.abd9941> PMID - 33536214). This study reports additional details on genetic maps, etc. In brief, the genetic map was calculated directly from the RIL genotype data (i.e. expansion could not be assessed). As part of our data checking, we corrected a single genotyping error that influences the apparent recombination frequencies. A depiction of the final genetic map, missing genotypes, and recombination frequencies are already provided in Supplementary Fig. 1. The genetic map is not perfectly even and this is partly because of large regions of strong conservation between the two closely-related wild strains (for example, on the left arm of chromosome V) where few SNPs exist. It is also likely influenced by uneven recombination frequencies across the genome.

Despite the previous publication, we have added the above summary to results and methods sections. In addition, more detailed information (e.g. genetic map) is provided in the Supplementary Information.

(5) Figure 3C. What is the significance line? Presumably GWER, but it would be good to specify in the Figure and Methods.

Response: The significance lines are the LOD thresholds, which were set to the 95th percentile of the maximum genome-wide LOD scores derived from 5000 random permutations of the data under the global null hypothesis of zero QTL. This was already stated in the Methods, but we have added an explanation to the figure legend as well.

(6) What is the broad-sense heritability for this trait? Also, why type of replication did the authors perform for the linkage mapping experiment? Please add these details to the Methods and Figure descriptions.

Response: We have added an estimate of the broad-sense heritability (H^2) of young adult PZ size to the Results (page 3). This was derived from an ANOVA ($PZSize \sim Strain + Env$) of the data in the Fig. 1d data set. Variance attributable to 'Strain' was divided by total variance. The ANOVA statistics and the H^2 calculation are now in Table S2D. For the linkage mapping experiment, RILs were divided into six scoring blocks and scored over the course of eight weeks. While some strains were measured in two separate blocks, not all were. Most strains were measured only once, with 30-40 individuals per strain. Averages and variances for strains that were measured twice were similar between measurements. Only one set of measurements was used for the QTL mapping. Additional detail has been added to the Methods section to clarify this.

(7) Figure 3D and G. Why not show the two-component QTL scan for the entire genome? I'm interested to see how the other chromosomes look.

Response: There are no other regions in the genome that are above the LOD threshold, so we feel this would be irrelevant.

(8) Figure 4A and B. What are the vertical ticks for each strain genotype?

Response: These are simply markers corresponding to 1MB, 3MB, etc. on chromosome V (as in black above) and are only included as an aid to the eye.

(9) Figure 5B. I think it would be more clear if the genetic backgrounds were plotted next to each other (e.g., JU200, JU1200 edit, JU751, JU751 edit).

Response: Excellent suggestion. These have been switched.

(10) The authors do a great job explaining the importance of their results and how epistasis can impact quantitative variation in the Discussion. In the second paragraph, it might be good to suggest that investigators can use the NIL backgrounds for allele replacements to avoid uncharacterized epistatic interactions from the genetic background. They have good evidence that this approach would help.

Response: We have added text to this effect in the Discussion.

(11) The authors should cite WormBase.

Response: We have corrected this oversight and cite WormBase in all relevant sections.

Reviewer #2 (Remarks to the Author):

(1) My one substantial concern is that the paper quickly dives into the minutiae of measuring PZ variation and its dependence upon genotype, without explaining whether there is any organismal consequence of the variation. For example, does PZ size correlate with total brood size, or egg laying rate? If so, there may be an adaptive angle on all of this. If not, we may instead be seeing "allowable slop" that is tolerated because the overall reproductive phenotype is not tightly connected to the PZ dynamics described here. Some sense of whether any of this ultimately matters to the worms would help motivate digestion of the many genetic details.

Response: We understand the reviewer's remark concerning the adaptive significance of variation in PZ size. Very deliberately, we aimed to avoid any simplistic explanations concerning the fitness consequences of variation in PZ size. While the germ cell progenitor pool is essential for reproduction to occur, it is less clear if quantitative variation in the number of progenitor cells

translates directly into fitness variation. We believe we have introduced this problem very carefully in the introduction of our paper and then discuss in detail the possible fitness links in two paragraphs of the discussion. In particular, we state:

“While activity of any germ stem cell system is obviously relevant for organismal reproductive fitness, it is not clear whether and how observed natural variation in germ cell proliferation (and PZ size) might translate into variation in reproductive fitness. Although increased germ cell proliferation has been found to correlate with increased egg-laying activity, offspring production, and egg quality^{10,25,31,32,58,88,89}, it is unclear to what degree this relationship is causal: *C. elegans* primarily reproduces through self-fertilizing hermaphrodites, which sequentially produce sperm, then oocytes for the remainder of life. In laboratory conditions, this causes *C. elegans* fecundity to be limited by the amount of self-sperm (~250) initially produced. PZ size is generally measured, as in our study, during the early adult stage, so that likely many of the observed progenitor cells will never develop into mature oocytes and become fertilized under selfing. However, increased germ cell proliferation allows for improved oocyte quality by upregulating the flux and number of oocytes undergoing physiological cell death (apoptosis), hence liberating resources to upregulate provisioning of surviving oocytes^{10,25,31,32,58,88,89}. Therefore, while adult PZ size need not reflect a direct proxy for future reproductive potential of selfing hermaphrodites (offspring number), it potentially increases resource provisioning of oocytes, thereby enhancing offspring quality. In addition, adult PZ size also reflects past proliferative activity of earlier larval stages, as illustrated by our quantifications of proliferative activity (Fig. 2B and 5D). Variation in adult PZ size may thus reflect differential reproductive investment during larval development, which may trade-off with energy allocation to somatic development.”

In addition, this discussion paragraph is explicit about the focal isolates of our study:

“With regard to reproductive fitness differences of the target isolates in our study, JU751 and JU1200, we have previously shown that selfing JU751 hermaphrodites have significantly reduced brood size relative to JU1200⁷¹. This reduction is not caused by differential sperm production but partly due to a major-effect variant causing early matricidal hatching in JU751⁷¹. Still, even after correcting for this genetic variant in JU751, brood size remains significantly smaller in JU751 relative to JU1200⁷¹. Possibly, reduced germ cell proliferation (and smaller adult PZ) in JU751 could thus reflect a lower reproductive investment. Of course, this scenario is highly speculative and observed differences need not be adaptive. Similarly, we ignore whether any of the detected QTL, including the *lag-2(cgb1007)* deletion, are maintained by selection. Even if selectively advantageous, the QTL variants could have been selected because of their effects on germline proliferative activity or unknown pleiotropic effects.”

(2) Summary, line 11: This was initially unclear to me. After reading the paper, I think an improved wording would be "Unexpectedly, restoring the deleted ancestral sequence in the isolate with a smaller PZ did not increase its PZ, but instead further reduced PZ size."

Response: The summary has been substantially revised to comply with the 150-word limit and now incorporates this suggestion.

(3) Page 4, section “Germ cell proliferative activity differs between wild isolates JU1200 and JU751”: Looking at Figure 1, one immediately wonders why the authors did not use the more extreme pairs, like JU1200 vs. CB4856, in their study. My guess is that it was because RILs already existed for these two strains. That is a good reason, but stating it explicitly here would help the reader better understand the choices made.

Response: We have added the word ‘existing’ to refer to the JU1200/JU751 RILs on Page 4.

Reviewer #3 (Remarks to the Author):

(1) To enhance data transparency, I advise providing the N numbers for PZ size analysis in all figures or explicitly referring to the supplementary dataset that contains the raw data. A good place for these would be Supplemental Tables already summarizing much statistical data.

Response: This information has been added to figures and/or figure legends throughout the manuscript. n values for each group have been added as a new column in Supplemental Tables 2b (Fig. 1d), 3c (Fig. 2a), 4b (Fig. 2b), 5b (Fig. 3a), 7b (Fig. 4c), 8b (Fig 4d), 9b (Fig 5b), 10b (Fig. 5d-f), 11b-e (Fig. 6) 12b (Supplementary Fig. 4). The n value for Fig. 1c was also added to Supplementary Table 1.

(2) Since *lag-2* is a signal involved in many developmental fates in addition to germ line stem and progenitor cell maintenance, it would be helpful if the authors commented on any extra-germline phenotypes consistent with a reduction in Notch signaling during larval development (for example, disruption of AC/VU cell fate specification, which is also thought to be dependent on HLH-2-mediated regulation). This would help establish whether the effects of the deletion are tissue-specific or affect multiple developmental events.

Response: Indeed, this is a relevant issue (although beyond the scope of our study in terms of experiments), which was only briefly addressed in the discussion of our initial submission. We have therefore made these statements more explicit in two sections of the discussion:

1. "Likewise, while we did not observe any obvious effects on other larval developmental processes involving Notch signaling (including regulation mediated by HLH-2) in strains carrying *lag-2(cgb1007)*, such as the AC/VU cell fate specification^{30,73}, we did not specifically search for them and cannot, therefore, rule them out."
2. "Even if selectively advantageous it remains unclear if QTL were selected because of their effects on germline proliferative activity or because of pleiotropic effects on other processes outside the germline, known to involve Notch signaling during larval development^{30,73}."